# ROS Homeostasis and Antioxidants in the Halophytic Plants and Seeds

**DOI:** 10.3390/plants12173023

**Published:** 2023-08-22

**Authors:** Hadi Pirasteh-Anosheh, Maryam Samadi, Seyed Abdolreza Kazemeini, Munir Ozturk, Agnieszka Ludwiczak, Agnieszka Piernik

**Affiliations:** 1National Salinity Research Center, Agricultural Research, Education and Extension Organization (AREEO), Yazd 8917357676, Iran; 2Natural Resources Department, Fars Agricultural and Natural Resources Research and Education Center, AREEO, Shiraz 7155863511, Iran; 3Department of Plant Production and Genetics, College of Agriculture, Shiraz University, Shiraz 7144165186, Iran; m.samadi@shirazu.ac.ir; 4Department of Botany and Centre for Environmental Studies, Ege University, Izmir 35100, Turkey; munirozturk@gmail.com; 5Department of Geobotany and Landscape Planning, Faculty of Biological and Veterinary Sciences, Nicolaus Copernicus University in Toruń, 87-100 Toruń, Poland; agnieszka.lud@umk.pl (A.L.); piernik@umk.pl (A.P.)

**Keywords:** oxidative, ROS scavenging, salinity, tolerance

## Abstract

Reactive oxygen species (ROS) are excited or partially reduced forms of atmospheric oxygen, which are continuously produced during aerobic metabolism like many physiochemical processes operating throughout seed life. Previously, it was believed that ROS are merely cytotoxic molecules, however, now it has been established that they perform numerous beneficial functions in plants including many critical roles in seed physiology. ROS facilitate seed germination via cell wall loosening, endosperm weakening, signaling, and decreasing abscisic acid (ABA) levels. Most of the existing knowledge about ROS homeostasis and functions is based on the seeds of common plants or model ones. There is little information about the role of ROS in the germination process of halophyte seeds. There are several definitions for halophytic plants, however, we believed “halophytes are plants that can grow in very saline environment and complete their life cycle by adopting various phenological, morphological and physiological mechanisms at canopy, plant, organelle and molecular scales”. Furthermore, mechanisms underlying ROS functions such as downstream targets, cross-talk with other molecules, and alternative routes are still obscure. The primary objective of this review is to decipher the mechanisms of ROS homeostasis in halophytes and dry seeds, as well as ROS flux in germinating seeds of halophytes.

## 1. Introduction

Approximately 2.5 billion years ago, reactive oxygen species (ROS) emerged on the planet alongside atmospheric oxygen, and have since played a critical part in the evolution of both prokaryotic and eukaryotic organisms [1,2]. Numerous studies have confirmed that ROS serves a plethora of advantageous and regulatory functions in various living organisms, including plants, animals, and other eukaryotes [1,3]. For instance, ROSs have been shown to play a crucial role in regulating different developmental processes, signaling stress, interacting with other organisms, eliciting systemic responses, and triggering programmed cell death in higher plants [3].

In an ideal situation, the coordinated activity of various antioxidant enzymes and compounds assists plant cells in balancing their levels of ROS [4]. Nevertheless, when stressors are present, the amount of ROS produced surpasses the intrinsic antioxidant defense system’s capacity to eliminate them, leading to oxidative harm to various cell components such as proteins, membrane lipids, and nucleic acids [5]. These deleterious effects of ROS are often termed oxidative stress.

One of the most common plant responses to salinity and drought stress is to reduce stomatal conductivity, thus minimizing water loss [6]. This response limits the Calvin cycle’s access to CO_2_ for carbon stabilization; consequently, the absorbed light exceeds the amount required for normal photosynthesis [7]. Combined with the toxic impacts of sodium and chlorine accumulated in the cytosol of plants under salinity stress, this excess excitation light by depleting electron receptors (QA in PSII and NADP in PSI) affects electron transfer through photosystems and also increases ROS production (mainly O_2_^•−^ and ^1^O_2_) by reducing O_2_ [8]. Limiting CO_2_ availability increases the rate of photorespiration in C_3_ plants, which in turn increases ROS production [9]. Therefore, salt-stressed plants use mechanisms to reduce ROS production that can maintain an acceptable level of net photosynthesis under CO_2_ limitation conditions and/or use alternative electron sinks that can prevent ROS formation from O_2_ [8].

Oxidative damage and the activity of antioxidant enzymes occur in halophyte plants like all plants. Halophytes are plants with very high tolerance to salinity stress that use different mechanisms for growth and survival in very saline environments. These plants grow in saline habitats in soils with a high salt concentration and with their special capabilities can replace ordinary plants in saline conditions [10]. Despite the differences in the definition of halophytes, in which plants are included, there are about 2500 to 3000 species of halophytes in the world [11].

It seems that halophyte species have a more noteworthy capacity than glycophytes to preserve net photosynthesis and to secure and stabilize both PS I and PS II beneath saline conditions [8,12]. Preserving net photosynthesis and protecting photosystems are the physiological mechanisms by which halophytes prevent the production of ROS in oxidative damage induced by salinity. Another interesting mechanism of halophytes to avoid oxidative damage is the switch from different modes of carbon assimilation; as some halophytes can change the carbon assimilation from C_3_ or C_4_ to Crassulacean acid metabolism (CAM) in high-salinity environments [13,14]. This switch is useful in reducing the production of salt-induced ROS in halophytes because CAM plants keep the stomata open at night and closed throughout the day to maintain photosynthesis during the day [8]. Although this mechanism is at first glance to prevent water loss during the day, however, by consuming the absorbed light in the current photosynthesis, it prevents the overproduction of ROS [15].

However, there is generally a dearth of knowledge about the production, scavenging, and roles of ROS in seeds and especially during the germination stage [16]. This information is even scarcer for the seeds of halophytes which naturally are salinity-tolerant plants of saline environments and hold immense potential to become non-conventional crops for arid saline lands in the future [5]. Therefore, this review aims to present an overview of the findings on (i) general mechanisms of ROS homeostasis in halophytes, (ii) ROS production and scavenging in dry seeds of halophytes, and (iii) ROS flux in germinating seeds under stress conditions.

## 2. Halophytes: Importance, Classification, and Salt Tolerance Mechanisms

The existence of huge resources of saline soil and water can be a major threat to sustainable agricultural production [2,15,17]. Globally, about 11% of the world’s irrigated lands are affected by varying levels of salinity [18]. However, many plants are sensitive to salinity, and their growth is limited even at low concentrations of salt [19,20].

A review of worldwide research into the introduction of salinity-tolerant cultivars shows that little success has been achieved in this regard. Thus, research on various crops in recent years has practically not led to actual salinity-tolerant cultivars for introduction to the farmer [21,22]. The findings of research on improving salinity tolerance using transgenic methods also show that a small number of these cultivars have been tested in the real field [19]. Therefore, the restrictions of common crops and the huge resources of saline water and soil have made the use of halophyte a suitable option for the production of fodder, edible seed, oilseed, biofuels, and green space inevitable [23]. Many halophyte species are used as forage; however, some are poisonous [24]. These taxa play an important role in the control of soil erosion. They are also used to clean contaminated soils and water [17,25].

Halophytes have the ability to naturally inhabit man-made areas like salt pans, roadside verges, and salt marshes [26]. This makes them ideal for bioenergy production and saline agriculture because they do not compete with crops for arable land or freshwater resources [27,28]. However, we must be cautious about the impact of activities on wild halophyte populations as their preservation is at risk due to exploitation. To ensure their long-term survival it is crucial to develop management plans that prevent the gathering of these valuable halophytic species [12,26,28].

Many studies have shown that at high salinity, the amount of halophyte production is much higher than the economic yield of commercial crops and trees [6,29]. However, it should be noted that the growth of the halophytes may be decreased at very high salinities. Therefore, is an inverse relationship between intensifying salinity and the proper halophyte number/abundance of halophytes [29], so only a few halophytes, like some *Salicornia* species, can grow with the salinity of seawater [22].

In general, the reaction of plants to salinity depends on their tolerance to soil salinity. Therefore, plants are divided into four categories based on the amount of dry matter production under saline conditions [30,31]:(1)Eu-halophytes: Growth of eu-halophytes is stimulated even in moderate salinities (such as *Salicornia europaea* and *Suaeda maritima*).(2)Facultative halophytes: Growth of these halophytes is slightly stimulated at low salinity (such as *Plantago maritima* and *Aster tripolium*).(3)Non-halophytes with low salinity tolerance: These plants are not halophytes and often include salt-tolerant crops and orchards. This group includes a wide range of economic plants such as barley (*Hordeum vulgare*), sorghum (*Sorghum bicolor*), cotton (*Gossypium* spp.), and pistachios (*Pistacia vera*).(4)Halophobic: Plants that are sensitive to salinity and even at low salinity levels there is a significant reduction in their growth and yield, such as saffron (*Crocus sativus*), common bean (*Phaseolus vulgaris*), and most vegetables.

Halophytes are categorized according to the mechanism of salt extrusion [32]:Recretohalophytes: Include halophytes that excrete salts on the outer surface (Exo-recretohalophytes) or the inside (Endo-recretohalophytes) of plant tissue.*Euhalophytes*: Or true halophytes with succulent leaves or stems.Pseudo-halophytes: Unreal halophytes that store salts in the parenchymal organs of the root.

By all categories, halophytes are plants with special capabilities that are good choices for the current global situation where severely limited freshwater resources, poor soil quality, and climate change have restricted the production of conventional plants [33]. Therefore, understanding the physiology of these species and elucidating their mechanisms for their high salinity tolerance is essential. Although good research has been carried out on the production of ROS, oxidative damage, and antioxidant enzymes in halophytes, more research is required to emphasize the role of ROS and antioxidants in the germination of halophytes.

Halophytes have various important mechanisms for salt tolerance, which work together to help them maintain ion balance and protect their cells from the effects of high salinity. The main important mechanisms of salt tolerance in halophytes include:Halophytes have developed ways to reduce the uptake of sodium ions (Na^+^) from the surrounding soil or water (reduction in Na^+^ influx). By preventing the accumulation of sodium ions in their cells, they can avoid salt-related damage [34,35].Halophytes possess compartments like vacuoles that can store and isolate excess sodium ions. This compartmentalization helps maintain sodium ion concentrations in the cytoplasm, which is crucial for cell health when dealing with high salinity conditions [34,35].Some halophytes have evolved salt glands or bladders that actively excrete sodium ions from their tissues (excretion of sodium ions.) By removing these ions from their cells, they effectively regulate salt concentration and protect themselves against saline stress [35].

These mechanisms collectively allow halophytes to tolerate and thrive in environments with levels of salt. By minimizing sodium influx compartmentalizing ions within structures and actively excreting them when necessary, halophytes ensure ion homeostasis and safeguard their cells from the damaging effects of elevated salinity [36].

## 3. ROS and Antioxidants in Halophytes and Their Role in Salinity Tolerance

Plants primarily generate ROS in their chloroplasts during photosynthesis, resulting in the production of O_2_•, H_2_O_2_, and O_2_^1^. Meanwhile, mitochondria produce O_2_• and H_2_O_2_ as a byproduct of respiration, and peroxisomes generate H_2_O_2_ during the process of photorespiration [4] as illustrated in Figure 1. Plant antioxidant defense systems include non-enzymatic and enzymatic systems involving compounds and enzymes that are distributed in different cellular compartments. The synergic action of both systems is responsible for the enhancement of the antioxidative response under salinity stress for many halophytes [37,38]. Enzymatic components of antioxidant defense include superoxide dismutases (SOD), catalases (CAT), peroxidases (POX) (glutathione peroxidase—GPX, ascorbate peroxidase—APX), and reductases (dehydroascorbate reductase—DHAR and monodehydroascorbate reductase—MDHAR). Certain antioxidant enzymes, including SOD and CAT, are speculated to have emerged as early as 3.6–4.1 billion years ago, prior to the great oxidation event that enabled organisms to cope with reactive oxygen species (ROS). These ROS emerged on Earth around 2.5 billion years ago, along with atmospheric oxygen [39]. The key non-enzymatic components are ascorbate, glutathione, tocopherol, phenolic, flavonoid, and carotenoid compounds [40,41].

By transforming O_2_• into H_2_O_2_, superoxide dismutase acts as “the first line of defense against ROSs”. Three primary varieties of SOD, namely cytosolic Cu-Zn SOD, mitochondrial Mn-SOD, and chloroplastic Fe-SOD, have been identified in plants [42]. Both glycophytes and halophytes exhibit a positive association between SOD activity and tolerance to salinity [8]. Nevertheless, halophytes are recognized to have relatively higher levels of SOD activity than glycophytes. For instance, when exposed to salt, *Cakile maritima* as a halophyte exhibits greater SOD activity than *Arabidopsis thaliana* as a glycophyte [43]. Yildiztugay et al. [44] indicated high SOD activity under toxic salt concentrations for *Salsola crassa*. However, after 30 days of salinity exposure, the activity of antioxidant enzymes in *S. crassa* was decreased. Therefore, the fast and stronger enhancement in SOD activity in halophytes could play an important role in stress signaling in halophytes. For example, SOD activity for halophyte *Tripolium pannonicum* can be dependent on salinity level and also on organs [45]. However, in *Salicornia europaea*, both ROS production and SOD activity are not growing at high NaCl concentrations i.e., 800 mM NaCl [46].

Catalases are enzymes consisting of 4 haem-containing subunits that help convert H_2_O_2_ into oxygen and water (H_2_O), effectively detoxifying it [47]. The CAT performs a range of functions, such as participating in photorespiration, eliminating H_2_O_2_ during the β-oxidation of fatty acids in germinating seeds, and promoting stress tolerance. Multiple isoforms of CAT are frequently present in plants, primarily located in the mitochondria or peroxisomes [48]. Catalases play a less significant role than SOD as was demonstrated for obligatory halophyte *S. europaea* [46]. Many other antioxidant enzymes such as glutathione peroxidase (GPX), glutathione S-transferases (GST), thiol peroxidase type II peroxiredoxin (Prx), and guaiacol peroxidase (GPOX) have also been reported from plants and contribute toward ROS homeostasis [8].

Peroxidases (POX) are glycoproteins catalyzing the oxidation of substrates by degradation of H_2_O_2_ analogical to CAT activity. The increase in the activity of POX and SOD at 300 mM NaCl in *S. europaea* was activated by the peroxidation of lipid membranes [49]. The highest peak intensities of POX activity were observed for *S. europaea* compared with *A. macrostachyum* and *S. fruticosa* from the same ecological habitat indicating the importance of POD activity in salinity tolerance strategy [50]. Kumar et al. [51] documented also that APX and POX are one of the main strategies for halophytes to control ion fluxes under high salinity for ten halophytic species.

Dehydroascorbate reductase (DHAR) and monodehydroascorbate reductase (MDHAR) are essential enzymes in the ascorbate-glutathione cycle, a key part of plant antioxidant defense mechanisms. DHAR and MDHAR regenerate strong antioxidant ascorbate utilizing dehydroascorbate (DHA) and monodehydroascorbate (MDHA) to maintain the pool of ascorbate in the cell [52]. The activity of both APX and MDHAR are enhanced in salt-stressed plants compared with unstressed plants. The addition of external ascorbic acid and tocopherol to plants affected by salt further heightened the activities of these enzymes compared with untreated salt-stressed controls [53]. DHAR and MDHAR activity was increased after long-term salinity stress for *S. crassa*. Additionally, the expression of these enzymes was found to increase 2–3 fold with increasing salinity in the halophytes *Urochondra setulosa* and *Dichanthium annulatum* [54].

To keep ROS levels within the tolerable range, plants also utilize low molecular weight non-enzymatic antioxidants such as ascorbate (AsA), glutathione (GSH), tocopherols, and flavonoids (Figure 2). Studies Anjum et al. [47] indicate a direct correlation between a plant’s tolerance to salinity and the levels of these antioxidants. AsA and GSH, two frequently occurring non-enzymatic antioxidants in plants, are present in all major compartments of plant cells, including the cytoplasm, apoplast, and chloroplast, with the ability to scavenge free radicals [55]. Their significance in enhancing salinity tolerance has been well documented. Ascorbate is a multifunctional cellular compound with an antioxidant and cofactor function. AsA is thought to play a role in mitigating the effects of salt stress by maintaining the osmotic balance in cells [56]. *Sphaerophysa kotschyana* as reported by Yildiztugay et al. [57] and *Limonium stocksii* as reported by Hameed et al. [58], both halophytic species, demonstrated an increase in AsA and GSH levels in response to salinity. Tocopherols, also known as vitamin E, are lipid-soluble compounds found in four distinct forms and are known to serve as an effective antioxidant defense for biological membranes [7].

The α-tocopherol, as a dominant form of vitamin E in the green tissues of plants, plays a crucial role in reducing the production of reactive oxygen species in the chloroplast under environmental stress conditions [59]. Ellouzi et al. [43] demonstrated an impressive antioxidant capacity of *Cakile maritima* correlated with the retention of a significant amount of α-tocopherol [49]. However, the level of α-tocopherol did not change significantly under salt stress, which implies that *Crithmum maritima* might neutralize the ROS production through direct quenching. The antioxidant role of phenolic compounds, including the subgroup flavonoids is particularly important in halophytes exposed to high salt conditions generating oxidative stress. The high content of flavonoids was noticed for *Salicornia europaea*, *Crithmum maritimum* L., *Mesembryanthemum edule*, and *Juncus acutus* [60].

The collaborative efforts of enzymatic and non-enzymatic antioxidants help maintain the levels of various ROS within a critical range necessary for regulating a variety of plant processes [8]. For example, ROSs are involved in the regulation of seed germination and dormancy [7], growth and development [61], stress acclimation, and programmed cell death [61]. However, these benefits are strictly dose-dependent. During periods of environmental stress, the production of ROS surpasses the cell’s ability to eliminate them, leading to elevated levels of ROS in the cell. This, in turn, results in oxidative damage to various cellular components such as nucleic acids, membrane lipids, and proteins [8]. Therefore, an effective antioxidant system is crucial for plants to cope with salinity stress, maintaining a crucial balance between the production of harmful ROS and their removal protects plant cells from potential oxidative damage. Furthermore, in response to salinity stress, some plants may even boost the production or activity of specific antioxidants to establish their ability to effectively manage this challenging condition [27,41].

## 4. ROS and Antioxidants in Dry Seeds

The seeds of many plant species have a remarkable ability to endure harsh environmental conditions such as high salinity and drought, as long as they are in a dry and inactive state. This means that most species’ seeds can remain viable in the soil for an extended period of time while still retaining the ability to sprout when conditions become favorable [62]. Even halophytes, which are plants that grow in salty environments, have seeds with low moisture content and can survive in a quiescent state. However, the production of ROS is a necessary part of seed development, germination, and seedling establishment. Therefore, an effective antioxidant system is critical for seed development, longevity, and the ability to germinate [16]. Hameed et al. [58] found that dry seeds of two halophytes, *Suaeda fruticose*, and *Limonium stocksii*, contain H_2_O_2_ and malondialdehyde. Seeds of *Cladium mariscus* L. (Pohl.) as a halophyte were studied for their nutritional and phytotherapeutic value. Antioxidants protect cells from oxidative stress by scavenging ROS, reactive species of nitrogen (RNS), and reactive species of sulfur (RSS) [63].

During the maturation and drying stage of seeds, ascorbate, and ascorbate peroxidase are typically depleted, which means that dry seeds usually lack these components. Nevertheless, there may be an ascorbate-independent antioxidant system in place, which could include enzymes such as SOD and CAT, as well as compounds like reduced GSH and tocopherols [64]. Hameed et al. [58] found that dry seeds of *S. fruticosa* and *L. stocksii* did not contain ascorbate, but they did have SOD, CAT, and glutathione reductase activity. The mechanisms by which quiescent seeds respond to ROS signaling pathways are still not fully understood, and it remains intriguing how seeds are able to sense ROS in their dry state.

The ascorbate-independent antioxidant system in dry seeds is a collection of enzymes and compounds that work together to protect the seeds from oxidative damage even in the absence of ascorbate and ascorbate peroxidase [47]. Superoxide dismutase is an enzyme that helps to neutralize superoxide radicals, which are highly reactive molecules that can cause damage to cell structures. Catalase is another enzyme that breaks down hydrogen peroxide, which is a type of ROS that can also cause damage to cells. Reduced glutathione (GSH) is a compound that can directly scavenge ROS and also act as a cofactor for other enzymes that play a role in antioxidant defense [47,58]. Tocopherols, also known as vitamin E, are a group of compounds that act as antioxidants by donating electrons to free radicals and thereby neutralizing them. Together, these components of the ascorbate-independent antioxidant system help to protect the dry seeds from oxidative damage and maintain their viability. However, it is still not entirely clear how this system works in detail, and further research is needed to fully understand its mechanisms [56,58].

The ascorbate-independent antioxidant system in dry seeds is thought to be an adaptation that allows seeds to withstand the oxidative stress that they may encounter during periods of desiccation and dormancy. During these periods, seeds are in a state of metabolic quiescence, which means that they are not actively growing or dividing. This quiescence is associated with a reduction in metabolic activity, including a decrease in the production of ROS [56,59]. However, while seeds are in this quiescent state, they are still exposed to environmental stresses that can generate ROS, such as exposure to light or fluctuations in temperature. The ascorbate-independent system is thought to provide a way for seeds to cope with these stresses and maintain their viability until conditions become favorable for germination [52,65].

One interesting aspect of the ascorbate-independent system is that it appears to be different in different plant species. For example, some species may have higher levels of SOD or CAT, while others may rely more on tocopherols or other compounds [7,64]. This suggests that the system has evolved independently in different lineages of plants and has been shaped by selective pressures in different environments. The ascorbate-independent antioxidant system is a fascinating adaptation that allows seeds to survive in harsh environments and maintain their viability over long periods of time. Ongoing research is helping to shed light on the details of how this system works and how it varies across different plant species [7,59].

While the ascorbate-independent antioxidant system in dry seeds has been extensively studied, many questions remain unanswered. For example, it is not yet clear how these components work together or how their activities are regulated during seed development, maturation, and dormancy. Additionally, the relative importance of each component in protecting seeds from oxidative damage may vary depending on the species and environmental conditions.

Furthermore, recent research has suggested that other compounds, such as polyamines and proline, may also play a role in the ascorbate-independent antioxidant system in dry seeds. Polyamines are organic compounds that can scavenge ROS and protect cell membranes, while proline is an amino acid that can act as a free radical scavenger and also help to stabilize proteins and membranes under stress conditions. The ascorbate-independent antioxidant system in dry seeds is a complex and dynamic network of enzymes and compounds that work together to protect the seeds from oxidative damage and maintain their viability. Further research is needed to fully understand the mechanisms underlying this system and how it varies across different species and environments.

## 5. ROS and Antioxidants in Germinating Seeds under Environmental Stresses

Abiotic stresses such as salinity, drought, heat, cold, heavy metals, high irradiance, and biotic stresses can all trigger the overproduction of reactive oxygen species (ROS), which can cause oxidative damage to critical cell components including membrane lipids, proteins, and nucleic acids [3]. In general, stress conditions can inhibit seed germination, and this inhibition is frequently associated with oxidative stress [66]. Therefore, the efficacy of an antioxidant defense system during germination is crucial for the successful germination of seeds. Lima et al. [67] reported that *Salicornia ramosissima* grown at moderate to high salinity levels (110 and 200 mM) indicated a higher antioxidant activity with higher levels of phenolic compounds. Al-Shamsi et al. [68] found that *Suaeda vermiculata* collected from highly saline marsh showed greater activity of CAT, GPX, and APX and less content of MDA and H_2_O_2_ as ROS. Duan et al. [69] also reported that salt stress enhanced the concentration of H_2_O_2_, and MDA, as well as, the activity of SOD and APX in the leaves of *Suaeda glauca* during both seedling emergence and seedling growth stages in two waterlogged and drained conditions. According to Wang et al. [70], after four days of exposure to high salinity (300 mM NaCl), the seeds of *Medicago sativa* and *Melilotus officinalis* had significantly higher levels of MDA than the control group, while the MDA content of germinating seeds of *Salsola drummondii* did not change with increasing salinity [71]. Meanwhile, exposure to salinity led to an increase of endogenous H_2_O_2_ and MDA in two halophytes, *Suaeda fruticose*, and *Limonium stocksii*, as well as in seedlings of *Sorghum bicolor*, as reported by Hameed et al. [58] and Chai et al. [72], respectively. 

The production of ROS and the response of halophyte seeds to oxidative damage appear to vary depending on the species and type of stress, but our current understanding is limited to a few studies. Therefore, more research is needed to fully elucidate these processes. In general, the activation of different antioxidant enzymes is often critical for successful seed germination, particularly under stress conditions [71]. For example, Pinheiro et al. [73] observed increased activity of SOD and CAT in germinating seeds of *Cucumis melo* after 48 h of exposure to NaCl. Similarly, the activity of SOD, CAT, and APX increased three days after sowing seeds of *Chenopodium quinoa* in various NaCl solutions, highlighting the importance of inducing antioxidant enzyme activity, particularly CAT, for successful seed germination under stress conditions [74]. Higher POX activity was noticed for halophytes *Plantago coronopus*, *P. maritima,* and *Basia sedoides* collected from different salinity areas [75] to effectively degradation of H_2_O_2_.

Our current understanding of the variations in the content of different non-enzymatic antioxidants in germinating seeds under stress conditions is limited. Hameed et al. [58] found that salinity inhibited seed germination in the coastal halophytes *Suaeda fruticosa* and *Limonium stocksii* by decreasing the ascorbate-dependent antioxidant system. Similarly, the concentration of ascorbate in germinating seeds of *Salsola drummondii* declined under high salinity and non-optimal temperatures [71]. In the absence of ascorbate, reduced glutathione serves as the key antioxidant in orthodox seeds [58]. However, environmental stresses can cause a decrease in its level. For example, reduced glutathione levels decreased with increasing salinity in germinating seeds of *L. stocksii*, *S. fruticose* [58], and *Melilotus officinalis* [70].

## 6. Conclusions

The production of ROS is a fundamental aspect of aerobic metabolism and is present throughout every stage of plant life, including seed development and germination. Despite their reputation as cytotoxic agents, ROS have emerged as crucial signaling molecules that regulate many plant processes, including seed germination and dormancy. The tight regulation of ROS levels creates an “oxidative window” that facilitates the germination of most seeds. The antioxidant defense system of seeds, comprising both enzymatic and non-enzymatic antioxidants, plays a pivotal role in this regulation. ROS contribute to seed germination by loosening the cell wall, weakening the endosperm, triggering signaling pathways, and reducing abscisic acid levels. This suggests that ROS and antioxidants have a more extensive role in seed biology than previously recognized, and further research is needed to clarify the downstream targets, cross-talk with other molecules, and alternative pathways involved in ROS function.

## Figures and Tables

**Figure 1 plants-12-03023-f001:**
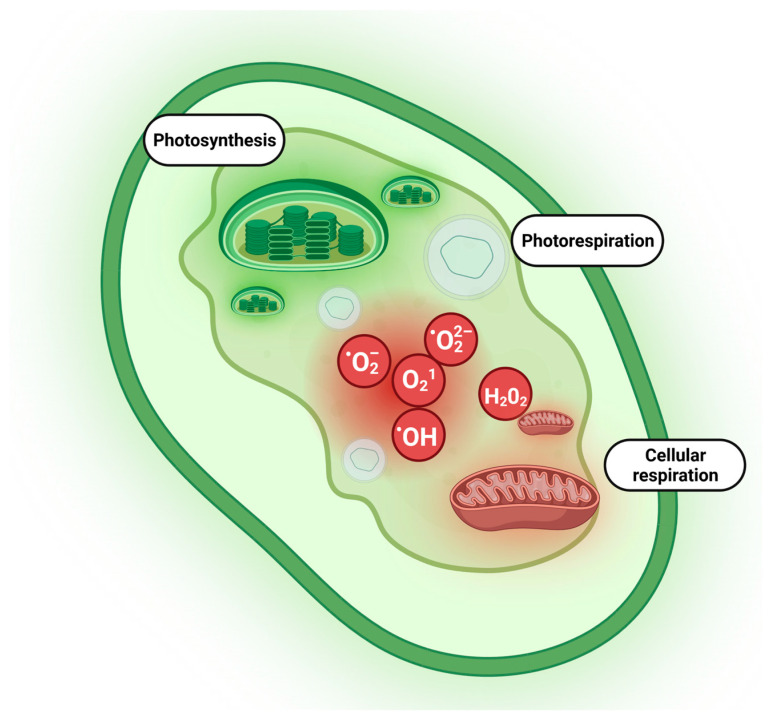
Major compartments of plant cells responsible for ROS generation. Photosynthesis, photorespiration, and cellular respiration as crucial processes responsible for ROS production were highlighted in the plant cell. ROS are constantly synthesized in chloroplasts, mitochondria, and peroxisomes as part of normal plant cell metabolism. The leak out in the electron transport chain in chloroplasts and mitochondria leads to the formation of superoxide (O_2_^−^) or hydrogen peroxide (H_2_O_2_). Oxidative metabolism in peroxisome produces H_2_O_2_, O_2_^−^, and singlet oxygen (^1^O_2_). Created with BioRender.com.

**Figure 2 plants-12-03023-f002:**
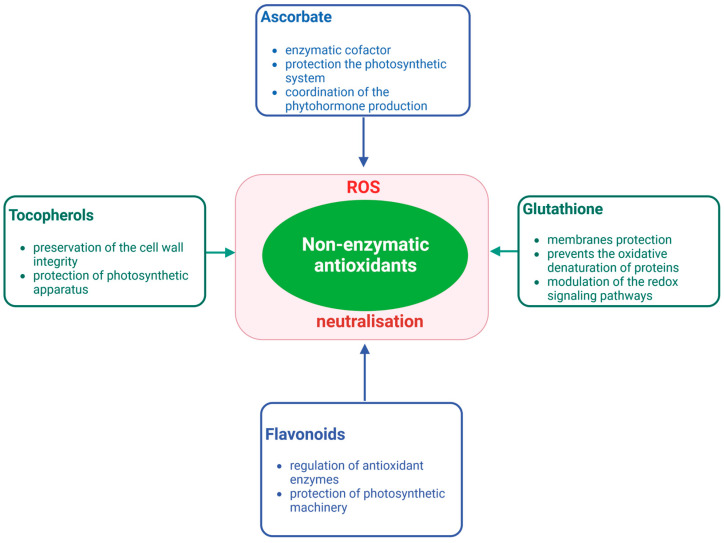
Essential non-enzymatic antioxidants and their multifunctional role. Ascorbate, glutathione, tocopherols, and flavonoids function as protection molecules against oxidative stress in the cell were marked. As non-enzymatic antioxidants cooperate to neutralize ROS and their negative impact on cells and by additional function (listed in the frame) mitigate abiotic stress. Created with BioRender.com.

## Data Availability

Not applicable.

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
