# Peer review of "ROS Homeostasis and Antioxidants in the Halophytic Plants and Seeds"

_plants, 2023, doi:10.3390/plants12173023_

Round 1

Reviewer 1 Report

The manuscript deals with a topic of interest and there are only a few review articles that refer specifically to halophyte seeds. The manuscript is appropriate for the scope of the special issue.

Overall, the text is well written and I have only a few minor observations. Firstly, in the title instead of the noun halophytes the adjective halophytic is more appropriate. This makes it clearer that the review refers specifically only to halophyte plants and their seeds.

Figure 1 is too general and not necessary. Figure 2 should be improved. In its current form it is misleading as it seems to indicate that each type of antioxidant is related to only one specific function. Please also revise the legend of this figure.

Author Response

I am grateful to the reviewer who provided valuable comments for the scientific improvement of the article. All his comments were fully applied as follows:

Comment: The manuscript deals with a topic of interest and there are only a few review articles that refer specifically to halophyte seeds. The manuscript is appropriate for the scope of the special issue.

Response: Thank you so much, your comments have richened the manuscript.

Comment: Overall, the text is well written and I have only a few minor observations. Firstly, in the title instead of the noun halophytes the adjective halophytic is more appropriate. This makes it clearer that the review refers specifically only to halophyte plants and their seeds.

Response: Based on this comment, we changed the title. The new title is: “ROS Homeostasis and Antioxidants in the Halophytic Plants and Seeds”

Comment: Figure 1 is too general and not necessary.

Response: Thanks to the reviewer, in this regard we changed the caption of Figure 1, so more detailed information is presented in this Figure.

Comment: Figure 2 should be improved. In its current form it is misleading as it seems to indicate that each type of antioxidant is related to only one specific function. Please also revise the legend of this figure.

Response: Great comment, we agreed with the reviewer, so we changed both the Figure and the caption. The Figure was re-drown in another form and its caption was re-written. The new caption for figure 2 is: “Figure 2. Essential non-enzymatic antioxidants and their multifunctional role. Ascorbate, glutathione, tocopherols, and flavonoids function as protection molecules against oxidative stress in the cell were marked. As non-enzymatic antioxidants cooperate to neutralize ROS and their negative impact on cells and by additional function (listed in the frame) mitigate abiotic stress. Created with BioRender.com“.

Reviewer 2 Report

This is a nice overview that deals with the production, scavenging, and roles of ROS and antioxidants in seeds, especially during their germination of halophytes. It is worth publishing.

Below I have given some suggestions: 

Give more details in the description of the Figure 1 (showing where these compartments responsible for ROS production are)

Line 94 Halophytes and their importance (try to give more details in the title)

line 111 what about halophytes that spontaneously colonize some anthropogenic sites  

line 113 Authors should complete suitable references here. 

line 194 ROS and antioxidants in halophytes and their role in salinity tolerance 

line 242 check the names of species, Latin names should be written in italics

line 282 I suggest changing the title ROS and Antioxidants in germinating seeds under environmental stresses 

Author Response

I am grateful to the reviewer who provided valuable comments for the scientific improvement of the article. All his comments were fully applied as follows:

Comment: Below I have given some suggestions:

Response: We applied all of your nice comments.

Comment: Give more details in the description of the Figure 1 (showing where these compartments responsible for ROS production are)

Response: The caption of Figure 1 was changed, so more detailed information is presented in this Figure.

Comment: Line 94 Halophytes and their importance (try to give more details in the title)

Response: Based on the opinion of the respected reviewer and according to the content of this section, we chose this new title for this section: 2. Halophytes: importance, classification and salt tolerance mechanisms

Comment: line 111 what about halophytes that spontaneously colonize some anthropogenic sites  

Response: The reviewer's interesting comment is appreciated. We have added the following to the article in this regard: Halophytes have the ability to naturally inhabit man made areas like salt pans, roadside verges and salt marshes (Polo-Ávila et al., 2022). This makes them ideal, for bioenergy production and saline agriculture because they don't compete with crops for arable land or freshwater resources (Hedayati-Firoozabadi et al., 2020; Fekete et al., 2022). However, we must be cautious about the impact of activities on wild halophyte populations as their preservation is at risk due, to exploitation. To ensure their long term survival it's crucial to develop management plans that prevent gathering of these valuable halophytic species (Polo-Ávila et al., 2022; Fekete et al., 2022; Pirasteh-Anosheh et al., 2022).

Comment: line 113 Authors should complete suitable references here. 

Response: Thanks to the accuracy of the reviewer's opinion, we have added reliable and relevant sources as follows: Many studies have shown that at high- salinity the amount of halophyte production is much higher than the economic yield of commercial crops and trees (Ranjbar et al., 2018; Pirasteh-Anosheh et al., 2023).

Comment: line 194 ROS and antioxidants in halophytes and their role in salinity tolerance 

Response: Based on this comment, we changed the title of that section as the reviewer suggested.

Comment: line 242 check the names of species, Latin names should be written in italics

Response: Thanks the reviewer, we checked the scientific names and corrected them.

Comment: line 282 I suggest changing the title ROS and Antioxidants in germinating seeds under environmental stresses

Response: Based on this comment, we changed the title of that section as the reviewer suggested.